# Clinical and Fundamental Research Progressions on Tumor-Infiltrating Lymphocytes Therapy in Cancer

**DOI:** 10.3390/vaccines13050521

**Published:** 2025-05-14

**Authors:** Jiandong Hu, Mengli Jin, Weihong Feng, Barbara Nassif-Rausseo, Alexandre Reuben, Chunhua Ma, Gregory Lizee, Fenge Li

**Affiliations:** 1Core Laboratory, Tianjin Beichen Hospital, Tianjin 300400, China; 18345189621@163.com (J.H.); 2120181338@mail.nankai.edu.cn (M.J.); 2Department of Oncology, Tianjin Beichen Hospital, Tianjin 300400, China; balala2005@163.com; 3Department of Melanoma Medical Oncology, The University of Texas MD Anderson Cancer Center, Houston, TX 77054, USA; bmnassif@mdanderson.org (B.N.-R.); areuben@mdanderson.org (A.R.); glizee@mdanderson.org (G.L.); 4Department of Immunology, The University of Texas MD Anderson Cancer Center, Houston, TX 77054, USA; 5Cancer Diagnosis and Treatment Center, Tianjin Union Medical Cancer Center (The First Affiliated Hospital of Nankai University), Tianjin 300121, China; mch8178@163.com

**Keywords:** tumor-infiltrating lymphocytes, molecular mechanisms, malignant solid tumor, clinical response, prognosis

## Abstract

Malignant tumors represent a significant threat to human health. Among the various therapeutic strategies available, cancer immunotherapy—encompassing adoptive cell transfer (ACT) and immune checkpoint blockade therapy—has emerged as a particularly promising approach following surgical resection, radiotherapy, chemotherapy, and molecular targeted therapies. This form of treatment elicits substantial antigen-specific immune responses, enhances or restores anti-tumor immunity, thereby facilitating the control and destruction of tumor cells, and yielding durable responses across a range of cancers, which can lead to the eradication of tumor lesions and the prevention of recurrence. Tumor-infiltrating lymphocytes (TILs), a subset of ACT, are characterized by their heterogeneity and are found within tumor tissues, where they play a crucial role in mediating host antigen-specific immune responses against tumors. This review aims to explore recent advancements in the understanding of TILs biology, their prognostic implications, and their predictive value in therapeutic contexts.

## 1. Introduction

Malignant neoplasms represent a considerable risk to human health. According to the Global Cancer Statistics, approximately 20 million new cancer cases and 9.7 million cancer-related fatalities were reported globally in 2022, with projections estimating around 35 million new cancer cases by the year 2050. The nine cancers with the highest incidence rates are predominantly solid tumors. Notably, China exhibits the highest incidence and mortality rates for cancer patients worldwide [1]. Of the 20 million newly diagnosed cancer cases, 4.82 million are attributed to China, and among the 9.7 million cancer deaths, 2.57 million occur in China [2].

The primary objective of immunotherapy is to mitigate endogenous autoimmune suppression through the utilization of immune checkpoint inhibitors (ICIs) or to enhance exogenous immune cell replenishment via adoptive cellular transfer (ACT) therapy. ACT encompasses various components, including tumor-infiltrating lymphocytes (TILs), T-cell receptor-engineered T-cells, chimeric antigen receptor T-cells, dendritic cells (DCs), and natural killer (NK) cells [3]. Within the tumor microenvironment (TME), the interaction between tumor cells and immune cells is often reciprocal; immune cells possess the capability to recognize and eliminate tumor cells, while tumor cells can evade destruction by establishing immunosuppressive signaling pathways. This interplay not only influences the initiation and progression of the disease but is also closely linked to patient prognosis [4].

In this review, we examine the advancements in research concerning TILs, focusing on their subpopulations, composition, prognostic implications, and predictive value. We also assess the clinical effectiveness of TILs in the treatment of various solid tumors, alongside the molecular mechanisms through which TILs facilitate anti-tumor responses. This compilation of data provides a thorough understanding of TIL-mediated immunotherapy and will serve as a foundation for future fundamental and clinical investigations into therapies related to TILs.

## 2. The Research Evolution of TILs Therapy

Steven Rosenberg’s group first used TILs to treat tumor-bearing mice and found that the therapeutic potency of TILs was 50–100 times higher than that of lymphokine-activated killer cells in 1986 [5]. In 1988, more studies reported that TILs have active roles in a variety of advanced malignant tumors [6]. In 2008, it was suggested that tertiary lymphoid structures (TLSs) are present in the tumor tissue. The formation of TLSs symbolizes the infiltration of immune cells and the enhancement of their tumor-killing effect, which is a positive factor for patient survival and prognosis, promoting tumor lymphocyte infiltration [7]. In 2015, LN-144 began clinical phase II trials to the end of the trial in 2021 [8]. In 2021, the FDA granted breakthrough therapy designation to TILs therapy LN-145 [9]. The FDA granted accelerated BLA approval for AMTAGVI™ on February 16, 2024 with a price of USD 515,000 per dose, which is the first TILs therapeutic regimen for advanced melanoma that has progressed following PD-1/PD-L1 therapy worldly. After 2020, China’s TILs injectable products gradually enter the clinical trial phase. In 2021, GT101 Injection from GRIT Biotechnology (CXSL2200061) first received implied clinical trial approval from the National Drug Administration (NMPA), GC101 TIL injection (CXSL2200070), and LM103 TILs (CXSL2300304) received implied clinical trials approvals from the NMPA subsequently. Excitingly, GT101, the fastest progressing TILs product in China, is expected to be on the market by the end of 2025 (Figure 1).

### 2.1. TILs Subtypes with Positive or Negative Regulation of the Immune Response

TILs are lymphocytes that have been isolated from tumor tissue and exhibit a high degree of heterogeneity among immune cell subpopulations. These subpopulations predominantly include CD8^+^ cytotoxic T lymphocytes, CD4^+^ T cells, dendritic cells (DCs), and natural killer T (NKT) cells, which are primarily associated with positive immune responses. Conversely, certain TILs, such as tumor-associated macrophages (TAMs), regulatory T cells (Tregs), and myeloid-derived suppressor cells (MDSCs), play a role in negatively regulating immune responses (Figure 2). These cells secrete a variety of cytokines and growth factors, including interleukin-6 (IL-6), IL-10, vascular endothelial growth factor (VEGF), and transforming growth factor-beta (TGF-β). The secretion of these factors inhibits the activation of DCs, CD8^+^ cytotoxic T lymphocytes, and NK cells, thereby facilitating tumor cells’ evasion of immune surveillance and contributing to tumor development, invasion, and metastasis [4]. A comprehensive review of the functions of TILs is presented as follows.

CD8^+^ cytotoxic T cells play a crucial role in the immune response by specifically recognizing endogenous antigenic peptide-MHC class I molecular complexes and subsequently inducing the death of tumor cells. Their cytotoxic activity is mediated through two primary mechanisms: the first involves the release of perforin, granzyme, and other cytotoxic substances that directly eliminate tumor cells. The second mechanism triggers apoptosis in target cells through the expression of the Fas ligand (FasL) or the secretion of tumor necrosis factor-alpha (TNF-α), which interact with surface receptors on the target cells [10].

CD4^+^ T-helper 1 (Th1) cells are activated upon interaction with peptide antigens presented by major histocompatibility complex (MHC II) molecules. These cells primarily secrete cytokines such as interferon-gamma (IFN-γ), TNF-α, and IL-2, which are instrumental in mediating cellular immunity. The cytokines produced by Th1 cells enhance the cytotoxic capabilities of NK cells and CD8^+^ cytotoxic T cells while simultaneously inhibiting the proliferation of Th2 cells [11].

CD4^+^ T-helper 2 (Th2) cells predominantly secrete cytokines including IL-4, IL-5, IL-6, IL-10, and IL-13, which are known to suppress anti-tumor immune responses. Research indicates that the infiltration of Th2 cells into tumor stroma is significantly greater than that of Th1 cells, resulting in a Th1/Th2 imbalance. This shift contributes to a state of immunosuppression that adversely affects the body’s anti-tumor immunity, ultimately facilitating tumor development and progression [12].

Regulatory T cells (Tregs) are typically characterized as thymus-derived CD4^+^ CD25^+^ FOXP3^+^ T cells, which play a crucial role in suppressing tumor-associated antigen-specific activation within tumor-draining lymph nodes. Tregs further infiltrate the TME to inhibit the functionality of effector cells and effector molecules. The primary mechanisms through which Tregs exert their inhibitory effects include (1) the secretion of soluble immunosuppressive molecules such as IL-35, IL-10, and TGF-β; (2) the high expression of high-affinity IL-2 receptors, which compete with activated T cells for IL-2, thereby affecting their survival; (3) the induction of apoptosis in CD8^+^ cytotoxic T cells and NK cells through granzyme A, granzyme B, and perforin-dependent cytotoxicity; and (4) the expression of CTLA-4 or the secretion of IL-35, which inhibits DCs maturation, diminishes their antigen-presenting capabilities, and promotes the generation of immunosuppressive DCs [13].

MDSC represent a significant component of the immunosuppressive tumor microenvironment (TIME), undermining host immune surveillance through various inhibitory mechanisms, including direct suppression of T cells, impairment of M1 macrophages, and upregulation of PD-L1 [14]. High infiltration of MDSCs is associated with poor prognosis in patients with cancer [15]. In esophageal squamous cell carcinoma (ESCC), the activation of MDSCs is regulated by IL-6 and other signaling pathways mediated by aldehyde dehydrogenase. Research indicates that MDSCs are heterogeneous, with CD38 potentially serving as a marker for MDSCs exhibiting enhanced immunosuppressive capabilities in esophageal cancer [16].

Different TIL subtypes have different effects on tumor development, which is related to their distribution in TME and the regulation of related signaling pathways. This part will be discussed in the subsequent content.

### 2.2. Bystander TILs in Cancer Immunology and Therapy

Recent studies indicate that a limited proportion of T cells are specific to cancer, while a significant number of T cells, referred to as “bystander” T cells, recognize antigens that are not associated with cancer [17]. These bystander TILs are characterized by their abundance, non-consumptive nature, specificity for common pathogens, and innate-like cytotoxic capabilities. This has prompted the consideration of bystander TILs as potential therapeutic targets in the development of novel treatments that aim to be more effective and less toxic [18]. Within the intratumoral CD8^+^ T cell population, T cells are categorized into tumor-reactive and bystander T cells based on their antigen specificity. Notably, tumor-reactive TILs are predominantly composed of CD103^+^CD39^+^ cells, whereas bystander TILs are primarily CD39^−^ cells [19]. An analysis of intratumoral CD8^+^ T cell response profiles in ovarian and colorectal cancers (CRCs) revealed that the ability to recognize autologous tumors was limited to approximately 10% [17].

The current application of bystander TILs presents novel avenues for tumor immunotherapy. Specifically, bystander CD8^+^ TILs that are reactive to various viral antigens have been identified within non-small cell lung cancer (NSCLC) tissues. Administration of IL-15 to NSCLC model mice previously infected with murine cytomegalovirus has been shown to selectively enhance the production of IFN-γ by these bystander CD8^+^ TILs, suggesting their potential utility in tumor immunotherapy [20]. Furthermore, a classification of tumor reactivity and bystander CD8^+^ TILs in treatment-naïve primary CRC patients, based on the expression of CD39 and CD103, indicated that DNA methylation plays a significant role in both tumor reactivity and the formation of bystander CD8^+^ TILs [21]. In conclusion, a thorough investigation into the molecular mechanisms that regulate the activation of bystander TILs, along with an evaluation of the functional contributions of these TIL subpopulations to tumor immunotherapy, is warranted [22].

### 2.3. T Cells Distribution Heterogeneously in Solid Tumors

Cancers are characterized by populations of cells exhibiting diverse molecular and phenotypic traits, a phenomenon referred to as intra-tumor heterogeneity (ITH). A study involving 11 localized lung adenocarcinomas (LUADs) conducted through multiregional whole exome sequencing revealed that all tumors displayed significant evidence of ITH [23]. Furthermore, the influence of neoantigen ITH on antitumor immunity was demonstrated, highlighting a correlation between clonal neoantigen load and overall survival (OS) in primary LUADs (*n* = 139). In early-stage NSCLC, CD8^+^ TILs responsive to clonal neoantigens were identified, exhibiting elevated levels of PD-1 in tumors enriched with clonal neoantigens [24]. Additionally, an analysis of metastases in a majority of melanoma patients undergoing targeted therapy or immune checkpoint blockade revealed considerable genomic and immunological heterogeneity, alongside significant variations in T-cell frequencies across all patients [25]. Notably, the majority of T-cell clones were localized to a single tumor region, indicating that distinct neoantigens present in different tumor regions may have contributed to the spatial variability observed in T-cell populations [26]. Collectively, these findings suggest that ITH in TILs holds potential clinical relevance and warrants further investigation into the associated regulatory mechanisms.

## 3. Predictive Value of Lymphocytes Infiltration in Malignant Solid Tumors

Studies have demonstrated that the density, subtype, and infiltration location of TILs dominate the prognostic judgment in a variety of cancer types [27], CRC [28], breast cancer [29], melanoma [30], and so on. In order to gain a more comprehensive understanding of the actual proportions of TILs across different cancer types, we aggregated single-cell sequencing data from various cancer tissues collected from 2020 to the present. We quantified the proportions of distinct TIL subtypes within cancer tissues at various clinical stages, as presented in Table 1. This analysis is expected to contribute positively to the understanding and investigation of the functional roles of TILs.

### 3.1. NSCLC

In 2020, a meta-analysis study was conducted including 11,448 patients. The results of the analysis showed that high density of TILs indicated favorable OS (HR = 0.80, 0.70–0.89) and PFS (HR = 0.73, 0.61–0.85) in patients with NSCLC; a high density of CD3^+^, CD4^+^, CD8^+^ TILs indicated high DSS and a high density of CD20^+^ TILs indicated favorable OS (HR = 0.65, 0.36–0.94). However, a high density of Foxp3^+^ TILs in the tumor stroma indicated lower recurrence/recurrence-free survival in NSCLC patients (HR = 1.90, 1.05–2.76) [42]. According to the meta-analysis which incorporated 15,829 NSCLC patients in 2024, the analysis indicated exhibiting TILs infiltration, demonstrating a significantly improved OS (HR: 0.67; 0.55–0.81). It was also observed that FOXP3^+^ was correlated with a poor OS (HR: 1.35; 0.87–2.11) [43].

### 3.2. CRC

A meta-analysis involving 5108 patients investigated the prognostic significance of the composition and localization of TILs in individuals diagnosed with CRC. The findings indicated that the presence of CD8^+^ and FOXP3^+^ T cells (excluding CD3^+^ T cells) was correlated with enhanced disease-free survival (DFS) and OS. Additionally, a high density of CD3^+^ T cells at the tumor’s infiltrating margins, along with increased intratumoral infiltration of CD8^+^ T cells, was linked to improved DFS [44]. CRC patients can be categorized into two distinct subpopulations based on microsatellite status: microsatellite instability (MSI) and microsatellite stability (MSS), as determined by microsatellite sequencing. The analysis revealed that CRC patients exhibiting MSI-high status with low TILs density had a poor DFS outcome [45].

### 3.3. Hepatocellular Carcinoma (HCC)

HCC represents the most prevalent form of primary liver cancer. A meta-analysis encompassing 46 studies with a total of 7905 HCC patients indicated that a higher density of intratumoral FOXP3^+^ TILs correlate with poorer prognostic outcomes for HCC patients. In contrast, patients exhibiting intratumoral or pericancerous infiltration of CD8^+^ TILs experienced a significant prolongation of OS [46]. In addition, data from 3541 patients in 20 papers were summarized. The results of the heterogeneity analysis were moderate to highly heterogeneous in all studies except for Tregs and neutrophils. High CD3^+^, CD8^+^, and NK cell infiltration predicted better survival (OS, DFS, and RFS; *p* < 0.05). Higher Treg and neutrophil infiltration predicted lower OS and DFS. There was no difference in survival between macrophages and B cells [47].

### 3.4. Cervical Cancer

Cervical cancer represents a significant threat to women’s health and mortality on a global scale. The human papillomavirus (HPV) infection is recognized as one of the predominant contributors to the development of cervical cancer [48]. A study involving 101 patients diagnosed with cervical cancer revealed that a high infiltration of CD66b^+^ neutrophils within the pericarcinoma and tumor stroma, as well as the presence of CD163^+^ macrophages in the pericarcinoma, were correlated with a reduced RFS [49]. The density of TILs was significantly higher in HPV-positive tumors that expressed p16 compared to those that were HPV-negative. An investigation of 148 patients indicated that a higher density of CD204^+^ M2 macrophages infiltrating the tumor was associated with a shorter DFS, which serves as a negative prognostic indicator for individuals diagnosed with cervical adenocarcinoma [50].

### 3.5. Breast Cancer

Breast cancer is clinically categorized into three primary subtypes: estrogen receptor/progesterone receptor (ER/PR)-positive, human epidermal growth factor receptor 2 (HER2)-positive, and triple-negative breast cancer (TNBC). The presence of TILs in breast cancer has been identified as a prognostic indicator that can enhance patient survival, particularly among individuals diagnosed with TNBC and HER2-positive breast cancer [51]. A study investigated 2148 samples of patients with early-stage TNBC shown that increased infiltration of TILs in the tumor stroma was significantly associated with prolonged DFS and OS [52]. In a cohort of 242 TNBC patients, those with T-bet^+^ tumors exhibited longer OS compared to their T-bet^−^ counterparts. Further, patients with CD8^+^ and T-bet^+^ had better RFS and OS compared to CD8^+^ T-bet^−^ tumors [53]. Furthermore, single-cell RNA sequencing conducted on 6311 T cells from breast cancer patients revealed a substantial presence of tissue-resident memory (TRM) CD8^+^ T cells, which expressed elevated levels of immune checkpoint molecules and effector proteins. The presence of these CD8^+^ TRM cells was significantly associated with enhanced survival outcomes in TNBC patients, indicating a more favorable prognosis [29].

### 3.6. Different Prognostic Value of TILs Across Various Cancer Types

In many cancer types, high CD8^+^ T cell infiltration is linked to a better prognosis, while high FOXP3^+^ Tregs infiltration may indicate a poor prognosis. This highlights their significant universal value as prognostic indicators in tumor immunology. However, there are differences across cancer types. For instance, in NSCLC, high CD20^+^ TIL infiltration also has prognostic significance, which is not always the case for B cell infiltration in other cancers. In breast cancer, TIL infiltration in T-bet^+^ tumors is associated with better survival, showing that specific T-cell subsets can have unique prognostic value in certain cancers. Moreover, in TNBC, the presence of TRM CD8^+^ T cells correlates with improved survival, possibly due to their effector functions and prolonged presence in the tumor microenvironment. Thus, TIL subtypes play different roles as the cancer type varies.

## 4. Clinical Efficacy of TILs in Treatment of Multiple Types of Malignant Tumors

In recent years, a total of 79 clinical trials investigating TILs therapy have been conducted between 2021 and 2024 (Figure 3). A geographic analysis reveals that 20.27% (21/79) of these trials are not currently recruiting participants, while 48.65% (44/79) are actively recruiting. Over the past four years, Phase 1 TIL products have constituted the majority of clinical trials. The data indicate that 61% (41/67) of these trials are located in China, with 24% (16/67) situated in the United States. In these clinical investigations, TIL products have predominantly been utilized for the treatment of solid tumors, with melanoma, NSCLC, and gynecological cancers being the three most frequently targeted malignancies.

### 4.1. Clinical Responses of TILs Monotherapy for Malignant Cancers

TILs therapy has shown significant clinical efficacy in several solid malignancies. The TILs product of LN-145 successfully treated patients with advanced cervical cancer in a phase II clinical trial with an ORR of 44%, including one CR, nine PR, and two unconfirmed PR. In response, FDA granted breakthrough therapy status of the LN-145 [9]. In 2021, Lovance Biotherapeutics successfully conducted a phase II trial of the TILs product (LN-144) to treat patients with advanced melanoma after ICIs progression. Patients received a mean number of TILs of 2.73 × 10^10^ with a disease control rate of 80%, ORR of 36%, CR of 3%, and PR of 33%. These results suggest that TILs therapy is a potential new treatment for patients with advanced melanoma whose disease has progressed after treatment with ICIs [8,54]. In 2024, two patients with metastatic cervical cancer were treated with autologous TILs. Patient #1 developed clinical PR after 6 weeks of TILs infusion and had a 33% tumor shrinkage at 12 weeks of follow-up. Patient #2 had stable disease at 6 weeks post-treatment. The results suggest that TILs infusion induces sustained and long-term systemic immune responses and reverses peripheral CD4^+^ T and CD8^+^ T percentages [55].

Moreover, neoantigen-specific TILs were studied to improve the anti-tumor response in the treatment of malignant tumors. In 2014, Tran et al. successfully expanded neoantigen-specific TILs in a patient with metastatic cholangiocarcinoma. The patient received an infusion of IL-2 and 42.4 billion TILs, which contained a high number of CD4^+^ neoantigen-specific T cells and achieved significant regression of liver and lung metastases [56]. In a phase II clinical trial (NCT01174121), a woman with metastatic CRC received a single infusion of 1.48 × 10^11^ TILs, approximately 75% of which were CD8^+^ T cells, which specifically recognized the KRAS G12D mutant. The patient achieved regression of all metastatic lesions [57]. Importantly, we summarized part of the TILs clinical trial data from 2021 to the present to visualize the TILs infusion experiments in tumor immunotherapy (Table 2) which implies that TILs therapy is getting more and more clinical response and tends to be the most promising immunotherapy for cancer.

### 4.2. Clinical Results of Combination Therapy of TILs and Other Immunotherapies

In order to further improve the clinical efficacy of TILs immunotherapy in solid tumors, researchers explored different mechanisms of TIL infiltration, proliferation, and target genes regulating TIL anti-tumor responses. Current studies mainly focus on TILs therapy combining ICIs [63], DC vaccines [59], and BRAF inhibitors [64].

For cancer therapy, ICIs may be used prior to tumor tissue resection or during the initial growth of TILs in order to achieve a more potent tumor-killing effect of infused TILs [58,61,62]. The combination of TILs and anti-PD-1/PD-L1 therapy has shown initial promising results in several recent clinical trials [58]. In a phase I trial (NCT03215810), patients with metastatic NSCLC whose disease had progressed after nivolumab treatment received infusions of TILs and IL-2, followed by nivolumab to enhance the durability of the TILs. Durable CR was achieved in 2 of 13 evaluable patients [58]. In 2018, a patient with hormone receptor-positive metastatic breast cancer experienced complete and durable tumor regression after treatment with TILs in combination with anti-PD-1 antibodies. The TILs they screened were CD4^+^ T cells predominantly (62.5%) that recognized mutants of four proteins (SLC3A2, KIAA0368, CADPS2, and CTSB) [65]. In 2022, 168 patients (86% treatment-refractory to anti-PD-1) treated with TILs or ibritumomab found that the median PFS in the TILs and ibritumomab groups were 7.2 months and 3.1 months, respectively; 49% and 21% of the patients achieved an OR. The median OS in the TILs and ibritumomab groups were 25.8 and 18.9 months, respectively. In patients with advanced melanoma, PFS was significantly longer in those treated with TILs than ibritumomab [60].

The DC vaccine can induce immune responses that activate and increase the number of TILs, and its combination with TILs therapy is also being evaluated in clinical trials (NCT01946373) [66]. The combination of TILs therapy with lysosomal viruses is also being explored because viruses can counteract tumor immunosuppression by producing cytokines that promote the anti-tumor effects of TILs [67]. Recently, it has been shown that DC activates CD4^+^/CD8^+^ T cells in lymph nodes as well as stimulates PD-L1^+^ lymph node-associated macrophages. Moreover, DC vaccines activate tumor-associated macrophages (TAMs). Thus, combining DC vaccines with PD-L1 blockade can achieve significant tumor regression by depleting PD-L1^+^ macrophages and inhibiting effector/stem memory T cells [68].

BRAF mutations are the most common mutation leading to hyperactivation of the MAPK pathway and are found in approximately half of all cutaneous melanomas. Activating BRAF mutations (V600E) induces immune escape mechanisms, giving cancer cells the ability to evade T cell immune responses [69,70]. The BRAF inhibitor vemurafenib has an ORR of up to 50% for the treatment of BRAFV600E mutant melanoma, which improves PFS and OS [71]. BRAF/MEK inhibitors promote the proliferation of melanoma-specific T cells in melanoma patients [72]. In metastatic melanoma treated with a combination of TILs and vemurafenib, 7 of 11 patients achieved an objective CR and two of them achieved a CR [73].

TMEs manipulations will also be a hot area of research in the near future, and some possible target molecules are as follows: (1) inhibitors of CXCR4, CXCL12, CCR4, and CCR2 to reduce inhibitory cellular infiltration [74]; (2) inhibitors targeting TAM CD47/SIRPα or TREM2 to reduce M2 cell migration [75,76]; (3) targets to maintain the balance of the CD96/CD226/TIGIT axis [77]; (4) drugs targeting IL-33/TAM/MMP-9 signaling to reduce the downregulation of NKG2D receptor [78]; and (5) Wnt pathway inhibitors [79]. Preclinical data from these experiments need to be verified by large-scale randomized clinical trials in the years to come. A variety of genetically engineered modifications of TILs, such as knockdown of PD-1 and increase of homeostatic cytokine expression, are still under ongoing clinical studies [80].

## 5. Molecular Mechanisms by Which TILs Play Anti-Tumor Response

TILs exhibit anti-tumor and proto-oncogenic properties, which are intertwined during tumor progression, forming a complex map of multiple signaling pathways (Figure 4). Investigating the roles and mechanisms of TILs in TME may help us better understand the patterns of cancer progression and identify innovative immunotherapy targets.

### 5.1. TGF-β Signaling Pathway

TGF-β is a pleiotropic cytokine with multiple immunosuppressive functions. The tumor itself, as well as the infiltrating cells of the TME, can serve as a source and target of TGF-β. Tumor-derived TGF-β is capable of limiting T cell infiltration or functionally blocking the differentiation of protective T lymphocyte populations [81,82]. In the advanced HCC stage, TGF-β promotes tumor progression by regulating immune cells such as Tregs, CTLs, TAMs, and NKs. Firstly, TGF-β can induce CD4^+^ CD25^−^ naive T cells to express FOXP3 through activation of SMAD-dependent pathways, mediating the formation of Tregs. Tregs in peripheral blood infiltrate into HCC tumor tissues and inhibit the anti-tumor effects of CTLs [83]. Secondly, TGFβ1 enhances antigen-induced PD-1 expression through Smad3-dependent and Smad2-dependent transcriptional activation in vitro T cells and in vivo TILs. The PD-1 subpopulation was higher in Smad3-deficient tumor-specific CD8^+^ TILs, which led to enhanced cytokine production in TILs and draining lymph nodes, resulting in increased anti-tumor activity [84]. Additionally, the TGFβ signaling in Treg cells took part in immunosuppressive functions in TME. A review summarized the central role for TGFβ’s function in Treg and Th17 cells, which is to balance the immune response in TME [85]. Additionally, in breast cancer and melanoma, Itgβ8 ablation in Treg cells impairs TGFβ signaling in intra-tumoral T lymphocytes, which strengthen the effector function of CD8^+^ TILs, leading to efficient control of tumor growth [86]. A recent study identified the anti-tumor effect of combination therapy with anti-PD-1 and lactate dehydrogenase inhibitors were more potent than that of anti-PD-1 alone. The reason for that is lactate could modulate Treg production by emulsifying Lys72 in MOESIN, thereby improving MOESIN interaction with transforming growth factor b (TGF-b) receptor I and downstream SMAD3 signaling. TGFβsignaling. Overall, TGFβ plays a role in the development of an immunosuppressive tumor microenvironment. On the one hand, it controls the expression of immune checkpoint proteins (e.g., galectin-9, VISTA, PD-L1, IDO1). On the other hand, it induces the differentiation of naive T cells into Tregs in the tumor microenvironment and inhibits the expression of granzyme B in cytotoxic T cells.

### 5.2. Wnt/β-Catenin Signaling Pathway

Wnt/β-catenin signaling is frequently stimulated in HCC, which primarily reduces the frequency and impairs the function of TILs [87]. Activation of Wnt/β-catenin signaling in HCC impairs the innate immune system by decreasing DC infiltration, which in turn impairs the adaptive immune response by decreasing migration of antigen-specific CD8^+^ T cells [88]. Wnt/β-catenin signaling also interferes with CTL action, leading to CTL exhaustion [89]. Pancreatic ductal adenocarcinoma is associated with activation of Wnt signaling. CD4^+^ TILs express Tcf7, encoding for the transcription factor Tcf1. Conditional inactivation of Tcf7 in pancreatic cancer CD4^+^ TILs resulted in an increase in CD8^+^ TILs and a decrease in Tregs in the TME [90].

### 5.3. NF-κB Signaling Pathway

Addition of agonistic anti-4-1BB antibody activates 4-1BB signaling in tumor tissues and accelerates the growth of melanoma antigen-specific memory CD8^+^ T cells by activating NF-κB signaling in CD8^+^ T cells. Simultaneously synergistic stimulation of 4-1BB also promotes DC maturation by activating NF-κB signaling in DCs. This suggests that the 4-1BB-mediated NF-κB signaling activation promotes the growth of TILs in early tumor fragment culture [91]. In recent studies, Hif1α deletion in NK cells leads to enrichment of the NF-κB pathway in tumor-infiltrating NK cells and promotes INF-1α expression, which leads to slower tumor growth and improved patient OS [92].

### 5.4. Ferroptosis Signaling Pathway

Ferroptosis is a form of cell death that differs from apoptosis. It is the result of iron-dependent accumulation of lipid peroxidation. INF-γ produced by TILs downregulates the expression of SLC3A2 and SLC7A11, impairs the uptake of cystine by tumor cells, and thus kills cancer cells through the induction of ferroptosis [93]. GPX4 is essential for maintaining the immunosuppressive function of Treg. GPX4 knockdown in Treg suppressed the function of Treg by increased intracellular lipid oxidation and ferroptosis [94]. In lung cancer, PD-1 signaling in intratumoral CD8^+^ T cells induced GATA1 binding to the promoter region of phospholipid phosphatase 1 (PLPP1), which inhibited the expression of PLPP1. Deletion of PLPP1 reduced the synthesis of phosphatidylcholine and phosphatidylethanolamine, ultimately promoting TILs death by ferroptosis and impaired antitumor immunity [95].

### 5.5. Apoptosis

Several death receptors (Fas, DR3, DR4, DR5, TNFR1) can trigger apoptosis upon activation by their respective ligands. Some cells in the TME including endothelial cells and MDSCs express FasL and trigger apoptosis of TILs. Thus, TILs apoptosis blocks anti-tumor immunity and interference of the Fas/FasL pathway in TME could improve the efficacy of cancer immunotherapy [96]. TiRP tumors have been documented to recruit and activate CD8^+^ TILs, and let them apoptose. Interference with the Fas/FasL axis prevents apoptosis in TILs. Polymorphonuclear MDSCs expressing high levels of FasL and enriched in TiRP tumors triggered TILs apoptosis. Blocking FasL improves the anti-tumor efficacy of T-cell therapy in TiRP tumors and improves the efficacy of checkpoint blockade in transplanted tumors [97]. A study showed that IL-10, as an immunosuppressive cytokine, prevented DC-mediated CD8^+^ TILs apoptosis through regulating IFN-γ production. This finding revealed a DC-regulating role of IL-10 to potentiate CD8^+^ T cell-mediated antitumor immunity [98].

### 5.6. PI3K-AKT Signaling Pathway

A study has shown that intermittent administration of the PI3Kα/β/δ inhibitor BAY1082439 in a PTEN gene-deficient prostate cancer model can overcome ICT resistance and unleash CD8^+^ T-cell-dependent anti-tumor immunity. BAY1082439 possesses preferential inhibitory Tregs activity and likely promotes the clonal expansion of CD8^+^ TILs. BAY1082439 converts cell-intrinsic immunosuppression into immunostimulation by promoting IFNα/IFNγ pathway activation, β2-microglobulin expression, and CXCL10/CCL5 secretion in cancer [99]. It has been shown that the interaction between CD155 and TIGIT in tumors disrupts the glucose metabolism of CD8^+^ T cells by inhibiting the activation of the PI3K/AKT/mTOR signaling pathway, which ultimately leads to a decrease in the cytokine production of CD8+ T cells as well as a decrease in the infiltration capacity of CD8^+^ T cells. In contrast, inhibition of CD155/TIGIT interaction restored CD8^+^ T cell capacity [100].

### 5.7. Hippo Signaling Pathway

YAP is a transcriptional coactivator of the Hippo signaling pathway that regulates organ size during development. Recent studies have shown that YAP is upregulated in Treg subpopulations and enhances FOXP3 expression and Treg function in vitro and in vivo [101]. YAP also plays an immunosuppressive role in CD8^+^ T cells. Loss of YAP in CD8^+^ TILs result in enhanced activation, differentiation, and function [102,103]. Therefore, YAP has great potential as a target for anticancer immunotherapy.

### 5.8. cGAS-STING Pathway

The cyclic GMP-AMP synthase (cGAS)–stimulator of interferon genes (STING) pathway mediates anti-microbial innate immunity by inducing the production of type-I interferons (IFNs) and inflammatory cytokines upon recognition of microbial DNA [104]. In recent years, the cGAS-STING pathway is gradually being studied in TILs of antitumor immunity. It was found that inhibition of FABP5 in Tregs triggered the release of mitochondrial DNA, leading to the activation of the cGAS-STING signaling pathway, which induced the massive production of IL-10 and promoted the inhibitory activity of Tregs, and ultimately facilitated immunosuppression in TME [105]. Recently, it has been suggested that in PDAC tumors, mitochondrial DNA in tumor cells induces the activation of the cGAS-STING signaling in tumor-associated macrophages, which promotes cellular secretion of INFα. INFα acts on macrophages, resulting in a high level of expression of BST2 which activates the ERK-CXCL7 pathway. After receiving CXCL7, CXCR2, the surface receptor of CD8^+^T, will activate the intracellular AKT-mTOR pathway, leading to the depletion of CD8^+^T cells, which will promote tumor development and poor prognosis [106].

The mechanisms of TILs are diverse and complex, involving antigen recognition and cytotoxicity, with various signaling pathways interacting to regulate TIL functions. TGF-β and Wnt/β-catenin signaling pathways mainly suppress TIL infiltration and function, promoting tumor immune evasion. In contrast, NF-κB, ferroptosis, and apoptosis mechanisms are crucial for regulating TIL survival, proliferation, and activation. PI3K-AKT, Hippo, and cGAS-STING signaling pathways also affect TIL immune functions at different levels. The synergy between these mechanisms is critical. For example, activating the NF-κB pathway during antigen recognition can enhance TIL cytotoxicity, while ferroptosis and apoptosis eliminate damaged or senescent TILs, maintaining the TIL population’s vitality. TGF-β can also influence the activity of other pathways like Wnt/β-catenin to jointly regulate TIL function.

Among these pathways, TGF-β and Wnt/β-catenin are key targets for addressing immunosuppression in the tumor microenvironment. Inhibiting these pathways may enhance TIL infiltration and function. Activating NF-κB and ferroptosis can directly boost TIL anti-tumor activity. Apoptosis and PI3K-AKT pathways require balanced regulation to maintain TIL population vitality while avoiding excessive loss of effective TILs. Hippo and cGAS-STING pathways, though less studied, offer new opportunities for modulating TIL function. Future research should focus on synergistically regulating these pathways to maximize the therapeutic potential of TILs. To further improve the anti-tumor immune response of TILs, targeting the above signals using genetic modification techniques including specific gene knockout via CRISPER cas9 technology or gene knock in technologies are rising rapidly.

## 6. Conclusions and Future Directions

TILs show great treatment potential in cancer therapy and demonstrate persistent anti-tumor responses in cancer patients. Nonetheless, TIL therapy faces many challenges in clinical use, including patient selection, cell preparation, expansion, and treatment standardization, all of which require greater precision. Moving forward, delving deeper into the biological makeup of TILs is essential, particularly in complex interactions within the tumor microenvironment. Gaining a better understanding of the molecular mechanisms, signaling pathways and collaboration among immune cell subtypes of TILs will shed light on their distinct behaviors across different tumors. This knowledge will be vital for refining treatment strategies. At the meantime, it is crucial to ramp up clinical research efforts by increasing sample sizes and encompassing a broader range of tumor types. This approach will play a key role in accurately evaluating the efficacy and safety of TIL therapy.

Furthermore, there are some problems in the clinical treatment of solid tumors with TILs: (1) Its treatment targets have some limitations. Patients with severe autoimmune diseases, history of organ transplantation, breastfeeding or pregnancy, allergy to cytokines, and other similar conditions are not suitable to receive TILs. (2) The preparation, amplification, and infusion of TILs in the AIT processes are complicated. Different research units do not have uniform standards in the preparation of different types of TILs, so the quantity and quality of TILs cannot be guaranteed. (3) The degree of infiltration of TILs varies in different solid tumors and clinical stages (described previously). Detection and characterization of the subpopulation composition, number, and regional distribution of TILs in tumors is important for finding more appropriate tumor-specific targets and for the prognosis of solid tumors. The future trend of clinical treatment is the combination of TILs adoptive therapy with conventional therapies (e.g., surgery and radiotherapy) and emerging therapies (ICIs and genetic modification). With the advancement of research techniques, more functional information of TILs will be discovered, interpreted, and applied, which will facilitate the further development of tumor immunotherapy with TILs.

## Figures and Tables

**Figure 1 vaccines-13-00521-f001:**
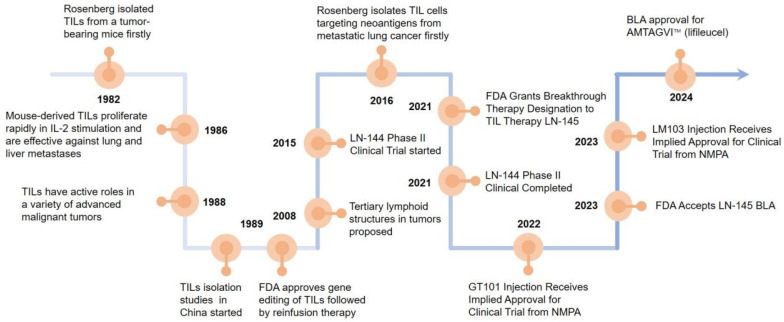
Landmark discoveries that aided the development of T cell-based ACTs. Key milestones in the development of T-cell-based adoptive cell therapies (ACTs) which starts with Rosenberg’s group demonstrating TILs’ high therapeutic potential in tumor-bearing mice in 1986. The recognition of TILs’ active roles in various advanced malignancies was addressed in 1988, the discovery of tertiary lymphoid structures (TLS) in tumor tissues was revealed in 2008, and the initiation of clinical phase II trials for LN-144 was conducted in 2015. The FDA granting breakthrough therapy designation to TIL therapy LN-145 in 2021 which accelerated BLA approval for AMTAGVI™ in 2024 are also marked. Finally, the entry of TIL injectable products in China into the clinical trial was recorded in 2020, with specific mentions of the clinical trial approvals for GT101 injection, GC101 TIL injection, and LM103 TILs from the National Drug Administration (NMPA).

**Figure 2 vaccines-13-00521-f002:**
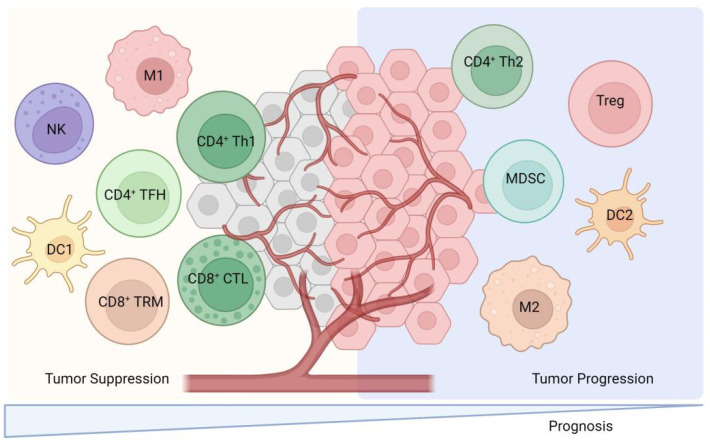
The cellular cross-talk between TILs subsets and their predominant contribution to either pro- or anti-tumor activities. This figure illustrates the diverse subsets of TILs and their roles within the tumor microenvironment. CD8^+^ cytotoxic T lymphocytes which recognize and eliminate tumor cells through mechanisms like releasing perforin and granzyme. CD4^+^ T cells are also depicted which include Th1 cells which enhance cellular immunity via IFN-γ secretion and Th2 cells that may suppress anti-tumor immune responses via IL-4 and IL-10 secretion. Regulatory T cells (Tregs) and myeloid-derived suppressor cells (MDSCs) are shown to negatively regulate immune responses by secreting factors like IL-10 and TGF-β. This figure also includes natural killer T (NKT) cells and dendritic cells (DCs) which contribute to the immune response against tumors. Th1: Helper CD4^+^ T. M: Macrophage. NK: Nature killer. TRM: Tissue-resident memory. TFH: Follicular helper CD4^+^ T. CTL: cytotoxic T lymphocyte.

**Figure 3 vaccines-13-00521-f003:**
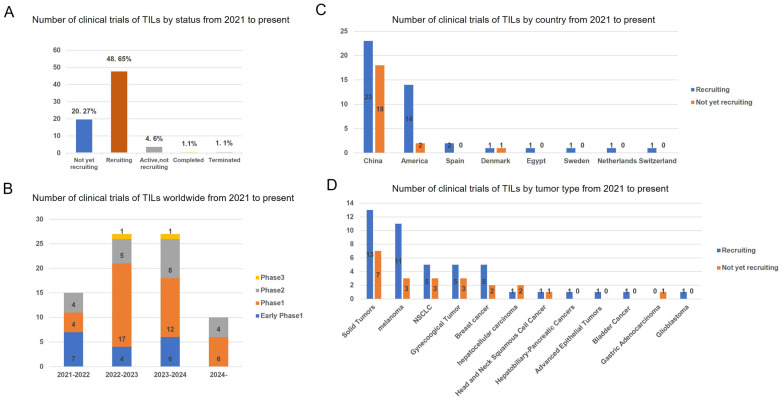
Status and regional disparities in clinical studies of TILs therapies. (**A**) The current status of all TIL clinical trials. (**B**) Number of clinical trials on TILs therapy worldwide by year and clinical stage. (**C**) Number of the geographic localization of TIL clinical trials worldwide. (**D**) Number of TIL clinical trials on different tumor type. The numbers of trials in each category followed by its percentage among total trials is shown. Data were obtained from Clinical Trials.gov.

**Figure 4 vaccines-13-00521-f004:**
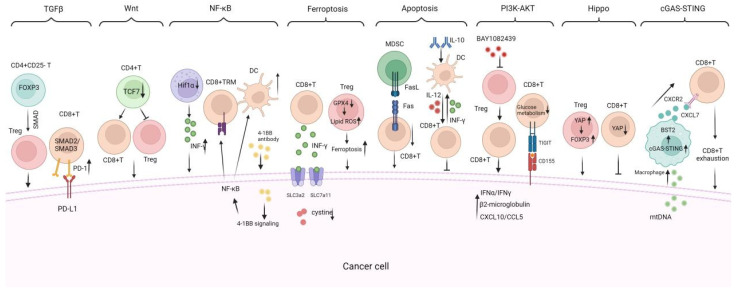
Critical signaling pathways regulating TILs infiltration, proliferation, and anti-tumor response. There are critical signaling pathways that regulate the infiltration, proliferation, and anti-tumor response of TILs including (1) the TGF-β signaling pathway, which suppresses TIL infiltration and function, promoting tumor immune evasion; (2) the Wnt/β-catenin signaling pathway which is activated in tumors that reduce DC infiltration and impair the migration of antigen-specific CD8+ T cells; (3) the NF-κB signaling pathway is shown to promote TILs growth and DC maturation; (4) the ferroptosis and apoptosis mechanisms, which regulate TILs survival and population vitality; (5) the PI3K-AKT, Hippo, and cGAS-STING signaling pathways are also included to illustrate their impacts on TIL immune functions. TRM: tissue-resident memory, TFH: follicular helper CD4^+^ T, BAY1082439: PI3Kα/β/δ inhibitor, GPX4: glutathione peroxidase 4; cGAS-STING: cyclic GMP-AMP synthase–stimulator of interferon gene. INF-γ: interferon-gamma. → indicates promote while ⊥ indicates inhibit.

**Table 1 vaccines-13-00521-t001:** The subtypes proportion of TILs in different types and clinical stages of cancer tissues.

Tumor Types	Cell Types	Subtype	Infiltration Rate	Total Cell Numbers	Total Patient Numbers	Clinical Stage of Tumor	Reference
Renal cell carcinoma	T	CD4^+^	62.75%	18,736	8	I	[31]
		CD8^+^	10.21%				
		Treg	27.04%				
GBM	T	CD4^+^	34.57%	N/A	11	IV	[32]
		CD8^+^	59.26%				
		Treg	4.32%				
IDH-G	T	CD4^+^	49.38%	N/A	15	II–IV	
		CD8^+^	40.12%				
		Treg	6.79%				
NSCLC	T	CD4^+^	32.50%	28,936	11	N/A	[33]
		CD8^+^	58.50%				
		Treg	5.73%				
NSCLC	TILs	CD4^+^	54.90%	N/A	150	I–IV	[34]
		CD8^+^	35.10%				
		Treg	1.89%				
		NK	0.82%				
Osteosarcoma	TILs	CD4^+^	20.13%	100,987	11	N/A	[35]
		CD8^+^	12.77%				
		Treg	10.51%				
		NK	5.24%				
		B	1.99%				
GBM	CD45+ immune cell	CD4^+^	8.75%	N/A	40	IV	[36]
		CD8^+^	5.83%				
		Treg	0.31%				
		NK	2.08%				
		B	2.08%				
		DC	83.00%				
		MDM	30.00%				
CRCLM	CD45+ immune cell	CD8^+^	26.52%	305,952	17	III–IV	[37]
		CD4^+^	37.82%				
		Treg	6.75%				
		Macrophage	11.23%				
		DC	43.95%				
		NK	8.06%				
		B	0.55%				
		Mast	0.39%				
		Monocyte	0.90%				
NSCLC [EGFR WT]	CD45+ immune cell	CD8^+^	31.64%	15,637	5	IA–IIIA	[38]
		CD4^+^	38.77%				
		B	5.07%				
		NKT	8.56%				
		Treg	10.94%				
		DC	0.97%				
		NK	1.87%				
NSCLC [EGFR MT]	CD45+ immune cell	CD8^+^	28.31%	16,250	5	IA–IIIA	
		CD4^+^	45.98%				
		B	3.06%				
		NKT	13.48%				
		Treg	7.51%				
		DC	0.10%				
		NK	1.21%				
Intrahepatic cholangiocarcinoma	Tumor	T	34.80%	31,275	8	II–III	[39]
		B	2.64%				
		NK	5.11%				
		Macrophage	10.89%				
		DC	2.45%				
Lung Adenocarcinoma	Tumor	T	49.00%	110,000	18	IA–IIIA	[40]
		B	8.00%				
		NK	4.00%				
		Treg	5.50%				
CRC	Tumor	T/NK	34.97%	20,412	7	I–IVA	[41]
		B	19%				
		DC	4%				
		Mast	1%				
Lung cancer	Tumor	T/NK	52.18%	54,197	8	IA3–IIIB	
		B	13%				
		DC	2%				
		Mast	1.79%				
Ovarian cancer	Tumor	T/NK	24.46%	20,642	5	IA–IVB	
		B	4%				
		DC	3%				
		Mast	0.31%				
Breast cancer	Tumor	T/NK	51.80%	27,789	16	II–III	
		B	10%				
		DC	1%				
		Mast	1.31%				

**Table 2 vaccines-13-00521-t002:** Partial ongoing clinical studies of TILs from 2021 to the present.

Trial	Phase	Tumor Types	Country	No of Prior Therapies %	Lymphodepletion IL2 Regimen	Efficacy Outcomes	Reference
NCT03215810	1	NSCLC	United States	Chemotherapy 25% Radiotherapy 10% None 50% Immunotherpy 15%	TIL infusion was defined as Day 0 as follows. Cyclophosphamide (60 mg/kg/day) for day 7 and 6 and fludarabine (30 mg/m^2^/day) for days 7 to 3 intravenously.	ORR: TIL (2/13)	[58]
NCT00338377	2	Melanoma	United States	N/A	Cyclophosphamide (60 mg/kg/day) for days 7 and 6 and fludarabine (25 mg/m^2^/day) for days 5 to 1 intravenously.	ORR: TILs +DC arm (4/8; 50%) TILs arm (3/10; 30%)	[59]
NCT02278887	3	Melanoma	United States	N/A	Cyclophosphamide (60 mg/kg/day) for 2 days and fludarabine (25 mg/m^2^/day) for 5 days intravenously.	PFS: 7.2(m), ORR: TILs (49%)	[60]
NCT04443296	1	Advanced cervical cancer	China	Chemotherapy 100% Radiotherapy 92.3%	N/A	ORR: 75% (9/12) PFS: 9–22 (m)	[61]
NCT01174121	2	Metastatic breast cancer	United States	N/A	Cyclophosphamide (60 mg/kg/day) for days 7 and 6 and fludarabine (25 mg/m^2^/day) for day 7 to 3.	CR: 16.6% (*n* = 6) PR: 33.3% (*n* = 6)	[62]

## Data Availability

Not applicable.

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
