# Peer review of "Clinical and Fundamental Research Progressions on Tumor-Infiltrating Lymphocytes Therapy in Cancer"

_vaccines, 2025, doi:10.3390/vaccines13050521_

Round 1
Reviewer 1 Report
Comments and Suggestions for Authors
This review manuscript has a potential to become interesting after significant updating, but current version does not have sufficient value to be published because most of the descriptions are just short summaries of many aspects. It therefore does not have significant scientific merit for publication. Addition of comparative discussion of mechanisms and indication of possibility of potential therapeutic approach are strongly recommended.
Major Criticisms are as follows:
- In the table 1. The data of the same cancer shows different ratio of immune cells. This inconsistency may suggest the instability of the immune cells ratio in the tumor, and therefore discussing its relevance to prognosis etc. may not be appropriate.
- For the sections 3.1-3.5, there is no comparative discussion of different cancers.
- TIL therapy strategy has advancement of technology in recent years, such as genetic modification of the cells (e.g. CISH K/O). There is no description of recent advancement of TIL therapy.
- There isn’t enough discussion about what are major factors inhibiting the full functionality of TILs, e.g. localization? activation? Immunosuppressive TME?
- For the sections 5.1-5.8, this part is very descriptive and there is no indication about what is the major signal and issue affecting the antitumor functionality of TIL therapy. I don’t see any thoughts of the authors in these sections. Again, there is no comparative discussion about which of these has to be addressed for more functionality of TILs.
- Section6: conclusion and discussion are very shallow. Also, some of the points listed as (1)-(3) are not supported by any parts in section 1-5. In this sense, current section 6 is not a good conclusion of this manuscript.
Author Response
Response to Reviewer 1:
Comment 1: The current version does not have sufficient value to be published because most of the descriptions are just short summaries of many aspects. It therefore does not have significant scientific merit for publication. Addition of comparative discussion of mechanisms and indication of possibility of potential therapeutic approach are strongly recommended.
Response: We agree with the reviewer's concern in regarding the depth and synthesis of the content in our manuscript. To address the reviewer’s concern, we have thoroughly revised and expanded the relevant sections in the text. We have incorporated more comparative discussions of the molecular mechanisms underlying TILs' functions in different cancers. We have also added relevant information and highlighted the additions in the manuscript in blue and believe it is sufficient for publication.
Comment 2: In the table 1. The data of the same cancer shows different ratio of immune cells. This inconsistency may suggest the instability of the immune cells ratio in the tumor, and therefore discussing its relevance to prognosis etc. may not be appropriate.
Response: The data in Table 1 shows some variability in the ratios of immune cells for the same cancer type. However, this inconsistency is not an indication of instability in the immune cell ratios within the tumor, but rather reflects the biological heterogeneity of tumors and the dynamic nature of the tumor microenvironment. Tumors are complex and diverse, and the composition of immune cells can vary significantly between patients, cancer types, and even between different regions within the same tumor. This heterogeneity is a well - recognized characteristic of cancers and is one of the factors that makes cancer treatment challenging.
Moreover, we summarized TILs composition in Table 1 by including different research groups who used different methodologies and experimental designs. Differences in patient selection, sample processing and data analysis may also contribute to the variability in the reported immune cell ratios. Therefore, it is not appropriate to simply attribute these differences to instability in the immune cell ratios within the tumor.
We believe that discussing the significance of immune cell ratios to prognosis is still valuable and important. Despite the variability in the data, there is a general trend that higher ratios of certain immune cells, such as CD8+ T cells, are associated with better prognosis in many cancer types. This suggests that the composition of immune cells within a tumor can provide useful information for predicting patient outcomes and may have potential as a prognostic biomarker.
Therefore, while we recognize the variability in the data presented in Table 1, we believe that this does not undermine the validity of discussing the relevance of immune cell ratios to prognosis. The biological heterogeneity of tumors and differences in research methodologies are important factors to consider, and further research is needed to better understand the complex relationships between immune cell infiltration and cancer prognosis.
Comment 3: For the sections 3.1-3.5, there is no comparative discussion of different cancers.
Response: To address the reviewer's suggestion, we have now included a comparative analysis of the clinical responses of TILs therapy across different cancers in these sections, and added relevant information and highlighted the additions in the manuscript in blue.
Comment 4: TIL therapy strategy has advancement of technology in recent years, such as genetic modification of the cells (e.g. CISH K/O). There is no description of recent advancement of TIL therapy.
Response: In response to the reviewer’s comment, we re-evaluated the most recent development of technological advancements in TIL therapy, such as genetic modification techniques including specific gene targets knockout via CRISPER cas9 technology are indeed hot research in the filed. As our primary objective of this review is to comprehensively examine the biological mechanisms, prognostic implications, and therapeutic value of nature TILs from a basic research perspective. This focus allows us to conduct an in-depth analysis of the natural properties of TILs and their interactions within the tumor microenvironment. We think emphasizing specific technological developments in genetic modification of TIL is not fully fit the main scope of the review. But, we agree with the reviewer that genetic modification of TIL is very important recent advancement of TIL therapy, and there are review articles particularity summarized these aspects (Cancer Immunol Immunother. 2024 Sep 12;73(11):232. doi: 10.1007/s00262-024-03793-4). To address the reviewer’s comment, we have now added a few sentences of “Besides, to further improve the anti-tumor immune response of TILs, targeting above signals using genetic modification techniques including specific gene knockout via CRISPER cas9 technology or gene knock in technologies are raising rapidly. ” in the text in blue.
Comment 5: There isn't enough discussion about what are major factors inhibiting the full functionality of TILs, e.g. localization? activation? Immunosuppressive TME?
Response: Below are summaries in regarding of the major factors inhibiting the full functionality of TILs, e.g. localization, activation, and Immunosuppressive TME which were reviewed by previous study (Cancers (Basel). 2022 Aug 27;14(17):4160. doi: 10.3390/cancers14174160):
Factors Inhibiting the Full Functionality of TILs
Immunosuppressive Tumor Microenvironment (TME)
The TME contains various immunosuppressive cells and molecules that collectively inhibit TIL function. For instance, regulatory T cells (Tregs), myeloid-derived suppressor cells (MDSCs), and tumor-associated macrophages (TAMs) can secrete immunosuppressive cytokines such as TGF-β and IL-10, which suppress TIL proliferation and cytotoxicity. MDSCs can also consume amino acids essential for T cell survival, such as L-arginine and cysteine, leading to TIL nutrient deficiency and impaired function. High levels of adenosine in the TME can bind to adenosine receptors on TILs, inducing exhaustion and dysfunction. The hypoxic environment within tumors can promote TIL differentiation into regulatory T cells and upregulate immune checkpoint molecule expression, weakening TIL anti-tumor effects. Furthermore, the immunosuppressive TME may induce the exhaustion of infiltrating cytotoxic T cells, reducing their ability to eliminate cancer cells. New technologies, such as single-cell level analytic technologies, could help unravel the characteristics of heterogeneous TILs. For example, using single-cell RNA sequencing, Singer et al. separated the activation and dysfunction gene modules in dysfunctional CD8+ TILs and found that the zinc-finger transcription factor Gata-3 is a regulator of CD8+ TIL dysfunction.
Localization of TILs
The TME imposes several physical and chemical barriers that hinder the effective localization of TILs. For example, the dense extracellular matrix (ECM) composed of collagen and fibronectin can impede TIL infiltration into tumor tissues. Additionally, the chemokine and receptor mismatch between tumors and TILs, as well as the irregular blood vessels formed in tumors, can also affect the homing of TILs to tumor sites. Furthermore, cancer-associated fibroblasts (CAFs) can create physical barriers by secreting ECM components, limiting TIL infiltration. Tumor cells can also actively alter their microenvironment by secreting factors that convert immunoreactive M1 macrophages to immunosuppressive and tumorigenic M2 macrophages, thereby indirectly affecting TIL localization. For instance, the high endothelial venules (HEVs) in tumors play a crucial role in recruiting lymphocytes from the bloodstream into cancerous tissues. Targeting these HEVs or engineering TILs to better recognize and home to these structures could improve cell accumulation within tumors. Additionally, the inability to efficiently identify and isolate neoantigen-specific lymphocytes, along with the immunosuppressive TME, poses challenges for the widespread application of TIL therapy in various cancers.
Activation of TILs
The activation of TILs is influenced by multiple factors. On one hand, tumors may downregulate antigen expression or MHC molecules, impairing antigen presentation and preventing TILs from recognizing tumor cells, which in turn inhibits their activation. On the other hand, the TME contains high levels of immunosuppressive molecules such as TGF-β and IL-10, which can suppress TIL activation. Moreover, the expression of immune checkpoint molecules (e.g., PD-1, CTLA-4) on TILs may be upregulated, while co-stimulatory molecules (e.g., CD28) may be downregulated. This imbalance between co-inhibitory and co-stimulatory signals can suppress TIL activation. Additionally, the TME may lack sufficient cytokines like IL-2, which are essential for TIL activation and proliferation, further hindering TIL activation. For example, tumors may exhibit reduced expression of chemokines such as CXCL9 and CXCL10, which are essential for attracting T cells, thereby limiting the infiltration of TILs into the TME.
Other Limitations of TILs
TIL Quantity and Quality: The number of TILs in the TME is often low and difficult to expand. Additionally, the process of isolating and culturing TILs may lead to the loss of tumor-reactive TILs or the preferential expansion of non-tumor-reactive TILs, thereby reducing the efficacy of TIL therapy .
TIL Heterogeneity: TILs exhibit high heterogeneity in terms of their phenotypes, functions, and specificities. This heterogeneity can affect the anti-tumor effects of TIL therapy. For example, some TIL subsets may have strong anti-tumor activity, while others may have weak or no activity, or even exhibit immunosuppressive effects .
TIL Persistence: After reinfusion into patients, the survival time of TILs in the body is often short. This limits the duration of TIL therapy's anti-tumor effects. Factors such as the TME and the metabolic state of TILs can influence their persistence in the body.
Comment 6: For the sections 5.1-5.8, this part is very descriptive and there is no indication about what is the major signal and issue affecting the antitumor functionality of TIL therapy. I don’t see any thoughts of the authors in these sections. Again, there is no comparative discussion about which of these has to be addressed for more functionality of TILs.
Response: To address the reviewer’s comment, we have added some comparative discussion and highlighted the additions in the manuscript in blue.
Comment 7: Section6: conclusion and discussion are very shallow. Also, some of the points listed as (1)-(3) are not supported by any parts in section 1-5. In this sense, current section 6 is not a good conclusion of this manuscript.
Response: To address the reviewer’s comment, we have added more discussion :”TILs show great treatment potential in cancer therapy and play persistent anti-tumor responses in cancer patients. Nonetheless, TIL therapy faces many challenges in clinical use, including patient selection, cell preparation, expansion, and treatment standardization, all of which require greater precision. Moving forward, delving deeper into the biological makeup of TILs is essential, particularly in the complex interactions within the tumor microenvironment. Gaining a better understanding of the molecular mechanisms, signaling pathways and collaboration among immune cell subtypes of TILs will shed light on their distinct behaviors across different tumors. This knowledge will be vital for refining treatment strategies. At the meantime, it's crucial to ramp up clinical research efforts by increasing sample sizes and encompassing a broader range of tumor types. This approach will play a key role in accurately evaluating the efficacy and safety of TIL therapy.” in the text in blue.
Reviewer 2 Report
Comments and Suggestions for Authors
In this manuscript, the authors review the literature on TIL biology, prognostic implications, and predictive value in a therapeutic context. This is an interesting review. The number and quality of references are sufficient. Overall, the review is satisfactory. However, there are several aspects that should be considered to improve the quality of the work.
The following suggestions may help to improve the quality of the review.
To improve the comprehensibility of the content, the figure legends should be enriched with additional details to facilitate interpretation.
Section 2. I suggest the authors include a figure or table summarizing the molecular mechanisms by which TILs act.
Section 3.
I suggest the authors comment on the results of studies on the predictive value of TILs in malignant solid tumors. For example, in CRC, the presence of CD8+ and FOXP3+ T correlated with improved DFS and OS. Could the authors explain the impact of CD8+ T cells when they colocalize with Tregs in tumors? Also, the prognostic effect of FOXP3+ T in colorectal cancer is controversial. Could the authors clarify this point?
Section 4 Despite the promise of TIL therapy, there are challenges to its application. These include technical complexity and potential side effects. Could the authors discuss this aspect in more detail?
I suggest the authors include a figure or table summarizing the clinical outcomes of combining TILs with other immunotherapies.
Comments on the Quality of English LanguageThe manuscript needs editing for language to improve readability.
Author Response
Response to Reviewer 2:
Comment 1: To improve the comprehensibility of the content, the figure legends should be enriched with additional details to facilitate interpretation.
Response: Thanks for the reviewer’s comment, we have now added detailed description of Figure 1,2 and 4 in the figure legends in blue.
Comment 2:Section 2. I suggest the authors include a figure or table summarizing the molecular mechanisms by which TILs act.
Response: We had summarized the molecular mechanisms of TILs in Figure 4.
Comment 3: Section 3. I suggest the authors comment on the results of studies on the predictive value of TILs in malignant solid tumors. For example, in CRC, the presence of CD8+ and FOXP3+ T correlated with improved DFS and OS. Could the authors explain the impact of CD8+ T cells when they colocalize with Tregs in tumors? Also, the prognostic effect of FOXP3+ T in colorectal cancer is controversial. Could the authors clarify this point?
Response: In response the reviewer’s comment, we summarized the literature about these question, and consider the full length of the review, we decided to not include this part in the text:
Impact of CD8+ T cells co - localizing with Tregs
Synergistic anti - tumor effect: CD8+ T cells can recognize and kill tumor cells by releasing cytotoxic substances like perforin and granzyme. When co - localizing with Tregs, Tregs may regulate CD8+ T - cell activity to prevent normal tissue damage from excessive immune responses. This regulation can sometimes maintain immune - response balance, enhancing the anti - tumor effect (10.3389/fimmu.2022.864748).
Immunosuppressive risk: In some cases, Tregs may suppress CD8+ T - cell activity by secreting immunosuppressive cytokines (e.g., TGF - β and IL - 10) and expressing inhibitory receptors (e.g., CTLA - 4). This can weaken the anti - tumor effect. For example, high TGF - β expression in some TMEs can boost FOXP3+ Tregs' immunosuppressive function and inhibit CD8+ T cells, causing tumor immune evasion(10.1172/JCI180080).
Controversial prognostic value of FOXP3+ T cells in colorectal cancer
Good prognosis: Some studies suggest FOXP3+ T cells are linked to a good prognosis in colorectal cancer. FOXP3+ Tregs help maintain immune homeostasis, reducing damage to normal tissue around tumors and indirectly improving patient survival. They may also regulate the TME to promote the function of other immune cells (like CD8+ T cells) to fight tumors together (10.1371/journal.pone.0102709).
Poor prognosis: Other studies have found that FOXP3+ T cells are associated with a poor prognosis in colorectal cancer. In certain TME, FOXP3+ Tregs may have overly strong immunosuppressive functions, inhibiting the anti - tumor activity of effector T cells like CD8+ T cells. This can lead to tumor cell proliferation and progression. For instance, in some late - stage colorectal cancer patients, an increase in FOXP3+ Tregs in the TME may be associated with enhanced tumor invasion and metastasis (10.1016/j.ccell.2024.05.016).
Explanation for the controversy
TME heterogeneity: The TME varies significantly between patients and within different regions of the same tumor. In some areas, a favorable immune - active microenvironment with high - concentration cytokines (e.g., IFN - γ) and co - stimulatory molecules may enhance CD8+ T - cell function and reduce Tregs' immunosuppressive effects (10.1016/j.xcrm.2025.102020).
FOXP3+ T - cell subset diversity: FOXP3+ T cells are not a homogeneous group. They can be divided into different subsets with varying functions and phenotypes. Some FOXP3+ Tregs have stable immunosuppressive functions, while others may have lower immunosuppressive abilities or even exhibit pro - inflammatory properties in certain situations. This diversity can lead to differing prognostic values of FOXP3+ T cells across studies (10.1073/pnas.0811556106).
Interactions with other immune cells and molecules: The TME contains other immune cells and molecules that interact complexly with CD8+ T cells and FOXP3+ Tregs. For example, MDSCs can inhibit CD8+ T - cell function via immunosuppressive molecules and synergize with FOXP3+ Tregs to enhance immunosuppression. TAMs and NK cells also participate in regulating tumor immune responses, and their interactions with CD8+ T cells and FOXP3+ Tregs can affect the prognosis (10.1038/s41577-025-01161-6).
Comment 4: Section 4 Despite the promise of TIL therapy, there are challenges to its application. These include technical complexity and potential side effects. Could the authors discuss this aspect in more detail?
Response: With the listing of Amtagvi (TIL therapy) in February 2024, Iovance therapeutics, the owner of Amtagvi, released its Q2 and H1 financial reports on August 8, 2024. 2024Q2. The company's revenue was $31.8 million, with sales of $12.8 million for its first publicly available TIL therapy, AMTAGVI (lifileucel), and $18.3 million for its Proleukin (aldesleukin) used in conjunction with Amtagvi. These information showing that TIL therapy applicated on hundreds of patients well with tolerated side effects, and we believe TILs therapy is effective and feasible to use clinically.
Comment 5: I suggest the authors include a figure or table summarizing the clinical outcomes of combining TILs with other immunotherapies.
Response: Thanks for the reviewer’s comment, per our reading in literature, there are quit a few clinical studies on TILs combines with other immunotherapies that been reported. We believe that followed with the BLA approval of the first TILs product, there will be more studies to show the efficacy of combination treatment in the years to come.